# The Relationship between Psychological Stress and Emotional State in Chinese University Students during COVID-19: The Moderating Role of Physical Exercise

**DOI:** 10.3390/healthcare11050695

**Published:** 2023-02-26

**Authors:** Baole Tao, Tianci Lu, Hanwen Chen, Jun Yan

**Affiliations:** College of Physical Education, Yangzhou University, Yangzhou 225127, China

**Keywords:** university students, physical exercise, emotional state, psychological stress, regulation

## Abstract

Objective: To explore the relationship between psychological stress and the emotional state of Chinese college students and the moderating effect of physical exercise. Methods: Students in a university in Jiangsu Province were randomly selected for the survey, and questionnaires were administered using the Physical Activity Rating Scale, the Profile of Mood States, and the Chinese College Student Psychological Stress Scale. A total of 715 questionnaires were distributed, and 494 valid questionnaires were returned. Among the students, there were 208 (42.1%) males and 286 (57.9%) females, with a mean age of 19.27 years (SD = 1.06). Results: We found a significant negative correlation between physical exercise and psychological stress (*r* = −0.637, *p* < 0.001); a significant negative correlation between physical exercise and emotional state (*r* = −0.032, *p* < 0.001); and a significant positive correlation between psychological stress and emotional state (*r* = 0.51, *p* < 0.001). Physical exercise negatively moderates the relationship between psychological stress and emotional state (*B* = −0.012, *p* < 0.01, Δ*R*^2^ = 0.007). Conclusion: Physical exercise is negatively correlated with both emotional state and psychological stress. Physical exercise can reduce the influence of psychological stress on emotional state and promote emotional health.

## 1. Introduction

From 2020 to 2023, the outbreak of COVID-19 led to a severe public health crisis in China. This epidemic is of great concern to the public. According to the Analysis of the results of the questionnaire on the impact of COVID-19 on residents’ lives and psychological well-being, 59.25% of the residents surveyed considered the impact of the epidemic on them to be “great”, 30.70% considered it to be “average”, and 9.9% considered it to be “very little”. Furthermore, 9.09% thought it was “minimal”, and 0.96% thought it had “no impact” [1]. Most people are constantly concerned about the epidemic, which prevents them from focusing on their everyday life, studies, and work activities. It makes them anxious and fearful about the epidemic and its effects. Suppose individuals are affected by the epidemic for an extended period. In such a case, they enter a state of stress, making them susceptible to physical illnesses caused by psychological emotions and affecting their physical health. They also suffer from varying degrees of anxiety due to the large amount of information they receive and process over a long period. It has also been reported that 66.9% of the public directly express varying degrees of anxiety, with 23.6% and 43.3% of the public saying they are “very anxious” and “relatively anxious”, respectively [2]. Based on the above, it is necessary to further explore the association between emotional states and stress disorders during the epidemic. During this period, psychological factors such as environmental adaptation, interpersonal communication, love, and employment are the primary forms of psychological stress that affect the healthy development of university students. Especially during the epidemic, university students may show anger, depression, resistance, or fear in response to stressful psychological events. The accumulation of long-term negative psychological states can quickly produce negative emotional states. Psychological research has found that emotions have an adaptive function and can keep an organism in an appropriate activity state, facilitating its adaptation to environmental changes. At the same time, emotions have a motivational function, which can drive the body to engage in certain activities and improve the efficiency of the activities. Among them, positive emotions coordinate and facilitate activities, while negative emotions play a disintegrating and destructive role [3]. The university student population is under high academic pressure and in a period of identity transition and adaptation to social role development, resulting in a high incidence of psychological stress. When an organism is in a state of chronic psychological stress-induced stress, its normal cognitive activities can be affected by all types of adverse emotional reactions (anxiety, depression, fear, etc.). Symptoms such as reduced mobility, reduced coordination of physical activities, and other behavioral maladjustments may also arise. In the long term, individuals may become addicted to unhealthy behaviors (smoking, compulsive drinking and eating, drug use, social withdrawal, etc.) and experience lowered self-esteem and a lack of social support, resulting in social maladjustment [4].

There has been increasing academic research on the stress psychology of university students under public health emergencies. During the early response to the SARS epidemic, university students’ anxiety status and coping styles were found to be closely related, with SARS having a highly significant impact on university students’ attention to class, emotional stability, and hygiene habits [5]. Out of extreme fear of sudden public health events, university students showed signs of inattention and emotional instability and increased concern for self-protection (personal hygiene, disease prevention, fitness, exercise, etc.). Similarly, during the epidemic, significant differences were found in the psychological state that university students showed, which were also regional, wherein southern students were found to experience higher psychological and physical burdens and uncomfortable physical reactions [6]. Another study found that the incidence of anxiety and depression among university students during the epidemic was 26.6% and 21.16%, respectively, and that students with a history of alcohol consumption and receiving more negative than positive information during the epidemic were more likely to have depressive moods [7]. Therefore, how to actively prevent and cope with the emotional impact of psychological stress on school students during the epidemic has become an urgent issue to be addressed by the government and universities.

The existing research has found that minor improvements in risk factors arising from psychological stress at an early age can have lifelong health benefits. Although university students are vulnerable to external stimuli, they are also more psychologically and behaviorally malleable, making psychological stress prevention and intervention at this time less costly and more beneficial. Existing research demonstrates that physical activity has a protective effect on physical and psychological health and a unique role in maintaining the stability of an individual’s emotional state under psychological stress. Physical exercise can cause physiological (enhanced adrenal activity) and psychological (regulation of emotional control and change of state of mind) effects in alleviating psychological stress, improving the individual’s ability to control psychological stress in different situations and atmospheres, and enhancing the individual’s social adaptation to cope with psychological stress. Regular participation in physical exercise improves the human central nervous system. It enhances the coordination of excitation and inhibition in the cerebral cortex such that the process of alternation between excitation and inhibition in the nervous system is strengthened [8]. This improves the balance and accuracy of the cortical nervous system, promotes the development of the body’s perceptual ability, and allows the brain to improve and increase the imagination’s flexibility, coordination, and reaction speed [9]. Furthermore, regular participation in sports also leads to the development of spatial and kinesthetic perception, making proprioception, gravity, touch, speed, and height perception more accurate, thus increasing the tolerance of brain cells to work. This demonstrates that physical exercise can increase general impedance resources and reduce the adverse effects of psychological stress. At the same time, physical exercise can buffer the psychological stress response, improve the cognitive function of the exerciser as a coping strategy, or be used as a preventive measure to make people’s responses to psychological stress stimuli more effective. In particular, physical exercise, as a positive coping modality, plays an essential role in individual emotional regulation and can effectively reduce the risk of adverse health states brought about by psychological stress. In today’s complex environment, individuals who regularly participate in physical activity can better adapt to adaptive emotional responses brought about by environmental changes, enhancing coping skills and contributing to emotional recovery [10]. Therefore, most scholars have concluded that physical exercise has anxiety-reducing and anti-stress effects [11,12] and can improve the physical self-esteem, self-identity, and self-efficacy of exercisers [13]. It can enable individuals to adopt more active and flexible coping strategies when dealing with daily problems, thus increasing positive coping styles and reducing negative coping styles [14].

Current research on the psychology of exercise has shown that the intensity of exercise influences the psychological effects of exercise. Physical exercise affects an individual’s emotional state in response to psychological stress. However, the effect of physical activity on mood is based on the interaction of physical activity intensity, duration, and frequency. Regarding exercise intensity and duration, regular physical activity is associated with less trait rumination and greater coping self-efficacy. However, strenuous exercise does not affect state rumination, perceived difficulty engaging in goal-directed behavior, or acquisition of post-stress emotion regulation strategies [15]. Brand et al. found that a 35-min exercise intervention increases individuals’ interest in social interaction and improves performance in facial emotion recognition [16]. Therefore, a moderate intensity of physical activity should be the preferred element. In terms of frequency of physical activity, although a single session of physical activity may affect how people respond or recover from emotionally stressful experiences, regular aerobic physical activity may produce resilience against prolonged or excessive emotional reactions [17]. Thus, both single and regular physical exercise can reduce exercisers’ reactions to stress, improve their ability to cope with or recover from stress, and help them more rapidly return to a stable emotional state. In summary, physical exercise affects the emotional state of psychologically stressed university students, where moderate physical exercise can improve subsequent emotional states such as anxiety, depression, tension, and fatigue, whereas high-intensity exercise may, on the contrary, increase negative emotions such as tension and anxiety [18,19,20]. Therefore, moderate physical exercise can change the intensity of one or more components of mixed emotions in real situations or adjust the relationship between different components through exercise, altering the nature of individual emotional states and thereby effectively regulating the impact of psychological stress on emotions in facilitating adaption to social requirements.

In summary, the results of existing studies demonstrate that psychological stress significantly affects the negative emotional state of college students and that physical exercise may have a moderating effect on such a relationship. Here, in this study, we propose the following hypotheses.

**H1.** 
*There is a positive correlation between psychological stress and college students’ emotional states.*


**H2.** 
*Physical exercise negatively modifies the relationship between psychological stress and emotional states in college students.*


Therefore, this study intends to examine the relationship between psychological stress and emotional state through a questionnaire study. In addition, we aim to examine the moderating relationship of physical exercise to the relationship between psychological stress and emotional state. It provides the basis for a scientific and rational understanding of the ability of physical exercise to enhance psychological regulation and protect the healthy development of emotions.

## 2. Materials and Methods

### 2.1. Research Subjects

For questionnaire distribution, a random sampling method was adopted to select survey subjects from Yangzhou University in Jiangsu Province. During this study, we randomly selected students at Yangzhou University to administer the questionnaire. Students from different classes were randomly selected to fill out the questionnaires each time, and the questionnaires were filled out in the classroom. The distribution period was from 22 February 2022 to 26 March 2022. A total of 715 questionnaires were collected in this study. After the questionnaires were collected, they were eliminated according to the following three criteria: (1) of all items, more than 30% were missing; (2) gender and age were not specified; (3) the items selected on the scale were the same. If the respondents all chose the same option, this indicates that the respondents did not answer the questionnaire carefully, and the questionnaire is, therefore, useless. After eliminating invalid questionnaires according to the above criteria, 494 questionnaires were finally selected, with an effective rate of 69%. Among them, 208 (42.1%) were male, and 286 (57.9%) were female. The subjects ranged in age from 17 to 21 years, with a mean age of 19.27 years (SD = 1.06). According to the formula
$$n=\frac{z^{2}\times p\left(1-p\right)}{d^{2}}$$
where *z* determines the confidence level, the *z* value we generally choose is 1.96, corresponding to a 95% confidence level; *p* is the percentage of a character in the target population. If there are no preliminary data, it is generally set to 0.5. Finally, *d* is the acceptable precision. We derived the minimum sample required for this study as 385. Therefore, 715 questionnaires were distributed in this study, which met the basic requirements. At the same time, according to the statistical perspective, at the required level of precision, 300–400 samples would have achieved a confidence level of 95% with a margin of error of no more than 5% of those randomly sampled. Therefore, the number of questionnaires distributed in this study met the requirements [21]. This study was reviewed and approved by the Ethics Review Committee of Yangzhou University (NO: YXYLL-2022-109).

### 2.2. Data Collection and Tools

Physical exercise was tested using the version of the Physical Activity Rating Scale (PARS-3) revised by Liang Deqing et al. [22]. This scale evaluates the amount of physical exercise regarding intensity, time, and frequency, such that exercise amount = intensity × time × frequency, where intensity and frequency are graded from 1 to 5 and correspond to a recording of 1–5 points, respectively, and time is graded from 1 to 5 grades corresponding to a recording of 0 to 4 points, respectively, with the highest possible overall score being 100 points and the lowest being 0 points. In this study, Cronbach’s α was 0.803.

The mood state scale (Profile of Mood States, POMS) compiled by McNair and revised by Zhu Belli et al. [23] was adopted. The scale contains seven subscales of tension, depression, anger, fatigue, panic, energy, and self-esteem (the first five subscales are of negative emotions, and the last two subscales are of positive emotions). It uses 40 words or phrases to describe emotional states, such as unhappy, nervous, carefree, energetic, tired, cheerful, nervous, etc., to evaluate the emotional state. The higher the score, the more negative the emotional state. In this study, Cronbach’s α was 0.879.

Psychological stress was measured based on the China College Student Psychological Stress Scale (CCSPSS) compiled by Liang Baoyong et al., which assesses the psychological stress level of college students over a period according to a total of 85 items. The five subscales are study, life, social interaction, development, and family. The 7-level scoring method of 1 (no influence or little influence)~7 (significant influence) was adopted. The higher the score, the higher the psychological stress level [24]. In this study, Cronbach’s α was 0.984.

### 2.3. Statistical Approach

Questionnaire tests were carried out using the IBM SPSS Statistics for Windows, Version 26.0. (Armonk, NY, USA: IBM Corp), and Cronbach’s α is the most commonly used test method for scale reliability. The value for Cronbach’s α coefficient ranges from 0 to 1, and the closer it is to 1, the better the reliability. Descriptive statistics and correlational analyses were conducted on college students’ psychological stress, emotional state, physical activity, and demographic information, and moderating effects were analyzed using the PROCESS program developed by Hayes et al., setting *p* < 0.05 as the threshold for statistically significant differences.

### 2.4. Quality of Reporting

We assessed the reporting quality of each included study using the Consensus-based Checklist for Reporting of Survey Studies (CROSS) [25]. The checklist consists of nineteen items, divided into six main categories. We calculated a percentage score with the underlying assumption that all criteria were weighted equally after excluding the criteria that were not applicable. Studies were assigned 1 point for reporting the item, 0.5 for partially reporting the item, and 0 for not reporting the item. Studies were categorized as ‘high quality’ if they met at least 75% of these standards, ‘moderate’ if they met between 50% and 75% of the relevant standards, and ‘low’ if less than 50% of standards were met. The quality assessment of this study reached 82.5%.

## 3. Results

### 3.1. Common Method Deviation Test

As the measures of the core variables in this study are derived from subjective judgment questions, the issue of homoscedasticity may exist. In this regard, the Harman one-way test was used to test the core variables for homoscedasticity. The results show that the variance explained by the most significant factor was 30.914% (less than 40%), so there was no severe problem of homophily bias in this study.

### 3.2. Descriptive Statistics of the Sample

The results of the descriptive statistics for each variable for the respondents in this study are shown in Table 1.

### 3.3. The Relationship between Physical Exercise, Emotional State, and Psychological Stress

Pearson product difference correlation analysis was conducted on physical exercise, emotional state, and psychological stress, with results shown in Table 2. The results show that there is a significant correlation between physical exercise, emotional state, and psychological stress, and there is a significant negative correlation between physical exercise and psychological stress (r = −0.637, *p* < 0.001). A significant negative correlation was found between physical exercise and emotional state (r = −0.032, *p* < 0.001), and there was a significant positive correlation between psychological stress and emotional state (r = 0.51, *p* < 0.001).

### 3.4. The Moderating Effect of Physical Exercise

In this study, the moderating effect was tested with psychological stress as the independent variable, emotional state as the dependent variable, and physical exercise as the moderating variable. The results showed that psychological stress had a positive effect on the emotional state (B = 1.3, *p* < 0.01), indicating that as the level of psychological stress increased, their emotional state became more negative; physical exercise harmed emotional state (B = −0.06, *p* < 0.01), indicating that as physical activity increased, their emotional state became more positive; there was a negative moderation of the effect of physical activity on psychological stress on the emotional state (B = −0.012, *p* < 0.01, ∆*R*^2^ = 0.007). Simple slope: When physical exercise was low, psychological stress had a strong positive effect on negative emotional states, which was more likely to lead to negative emotional states (Table 3). When the level of physical exercise was higher, psychological stress had a lower positive effect on negative emotional states, indicating that the influence of psychological stress on negative emotional states was weakened (Figure 1).

## 4. Discussion

### 4.1. Relationship between Psychological Stress and Emotional State

In this study, the relationship between physical exercise, emotional state, and psychological stress in college students was correlated through a questionnaire survey, and the results show there is a significant positive correlation between emotional state and psychological stress level and a significant negative correlation between physical exercise, emotional state, and psychological stress level. These results are also consistent with the findings of Yan Jun et al. [26]. Physical exercise reduces negative emotional states and effectively antagonizes psychological stress. The group in this study was the same as the one mentioned above, and the results obtained are consistent. We, therefore, hypothesize that there is a stable relationship between physical activity, psychological stress, and negative emotional states in the university student population. Psychological stress is a specific psychological and physiological response that occurs when an environmental stimulus, such as a stressor or stress response, threatens a person’s critical needs and ability to cope, or it can refer to the state of physical and mental tension that occurs in response to an environmental stimulus. Psychological stress, as an adaptive defense process, is often associated with highly charged emotional states [27]. It can inhibit or disrupt a person’s activities, or it may cause them to become active and their thoughts to become clear and explicit. On the emotional side, the positive side of psychological stress is that it can sharpen one’s emotional experience. However, when psychological stress is excessive or prolonged, it can lead to adverse emotions, such as tension, fear, anger, depression, and, in severe cases, even panic, loss of emotional control, crying, and shouting. Moreover, in terms of thinking, excessive psychological stress can cause one’s attention to become narrowed such that selective attention and thought regurgitation occur [28]. As the subjects of this study are university students in school, there was a severe lack of social interaction with social groups because of the epidemic. Therefore, prolonged psychological stress may lead to negative emotions and is a significant risk factor for anxiety and depressive episodes during individual development.

### 4.2. The Moderating Effect of Physical Exercise

Physical exercise is a critical intervention to buffer the effects of psychological stress on emotional states. In psychological stress research, several demographic and biological factors influence the intensity of stress levels. Due to their biological and genetic factors and the different natural, human, and social environments around them, individuals may react differently to the same stressor. Physical exercise is a somatic stressor that involves a stress process that occurs when changes in the internal and external environment stimulate the body [29]. The “trans-stressor adaptation hypothesis” explains why physical exercise, as a somatic stressor, can regulate psychological stress levels. This can lead to a series of beneficial adjustments in the motor, circulatory, respiratory, neuroendocrine–immune, and other systems, which can increase the ability to adapt to other stressors, including social stressors, through both psychological and physiological means [30]. It has also been shown that physical exercise allows the body to reflexively establish adaptive and anti-stress self-protection mechanisms, increasing the individual’s ability to function in high-stress situations, generally, and reducing anxiety and depression. For example, aerobic exercise (rhythmic gymnastics, walking and running exercises, swimming, cycling, etc.) can help reduce the physiological response to stress and speed up the recovery process by improving the function of the cardiovascular system, slowing down the heart rate, lowering blood pressure, and enhancing pulse output and oxygen uptake [31]. Furthermore, physical exercise can modulate mood and improve cognition, thereby buffering psychological stress levels, and exercise is associated with blunted physiological stress responses to prolonged physical activity, increased levels of brain-derived neurotrophic factor (BDNF) [32], and improved autonomic nervous system function [33].

Furthermore, Damasio’s somatic marker hypothesis also explains the effect of physical exercise on emotional states by modulating psychological stress [34]. When people experience stressful events, the physical changes that mark such stressful reactions form an “emotional-feeling loop”, with pleasant or unpleasant feelings generated by these changes constituting a somatic marker of emotional feelings in response to the particular stressful event, wherein physical exercise is also a form of stress. However, the physical and psychological feelings it brings to the individual differ from the negative ‘emotional-feeling loop’ caused by psychological stress, as physical exercise brings positive emotional experiences that promote the physical and mental health of the exerciser. At the same time, physical exercise may improve psychological stress through both psychological (coping styles, coping efficacy, and defense mechanisms) and physiological (neurological, endocrine, and immune function resources) pathways in order to restore internal stability and psychological well-being [26].

At the same time, the moderating effect of physical activity on negative emotions influenced by psychological stress in university students is also primarily related to the positive affective experience brought about by physical activity. Among other things, the sense of fluidity brought about by physical exercise is an ideal internal state of experience for humans [35]. In this state, the individual forgets the self and fully engages in the activity. The experience of the process is fun and enjoyable and creates a sense of control over the exercise process. A sense of fluency is, therefore, a positive emotional state. On the other hand, when an individual is in a negative mood brought on by psychological stress, a certain amount of physiopsychological energy builds up within the individual’s body and mind. This energy needs to be released through normal channels; otherwise, the individual will release it in inappropriate and excessive behavior, causing damage to society, families, and individuals. Physical exercise causes muscular exertion, profuse sweating, and a shift in attention, which allows excessive physiological energy to “evaporate”, relieving excessive psychological tension and depleting the negative psychological energy generated by stress [36]. Thus, physical exercise can be a coping strategy to reduce physiological and psychological stress and a preventive measure to alleviate the negative emotional impact of socially negative stress behaviors. Therefore, physical exercise can regulate the negative emotional state produced by psychological stress and contribute to the emotional well-being of individuals.

### 4.3. The Limitations of the Study

Psychological stress is an essential issue in psychological research. Individuals undergo a range of cognitive, emotional, physiological, and behavioral responses to stimuli from stressors. In the present study, we have found that physical exercise can regulate emotional states under psychological stress. However, the effects of physical exercise on psychological stress on emotional states are complex, and findings from different studies are inconsistent. In the present study, a cross-sectional approach was adopted, with which causal relationships cannot be revealed. Future research in the form of experiments is thus still needed to investigate causality. Moreover, the validity of this study may need to be revised due to factors such as sample sampling. Therefore, the sample size should be increased, and the scope of the respondents expanded in future research. Finally, this study group is composed of college students, which cannot reflect the emotional state of all people. In order to make the study more valuable, future surveys of national subjects are needed.

## 5. Conclusions

This study found that physical exercise was negatively associated with emotional state and psychological stress among Chinese college students during the epidemic. In contrast, a positive association was found between psychological stress and emotional state. This shows that the higher level of psychological stress, the more negative the emotional state of Chinese college students in the context of the repeated persistence of the epidemic. Physical exercise can act as a protective factor to reduce the effect of psychological stress on emotional state and prevent it from turning negative. Therefore, college students need to engage in physical exercise to stabilize their emotional states when the epidemic is in the rising outbreak and control periods.

## Figures and Tables

**Figure 1 healthcare-11-00695-f001:**
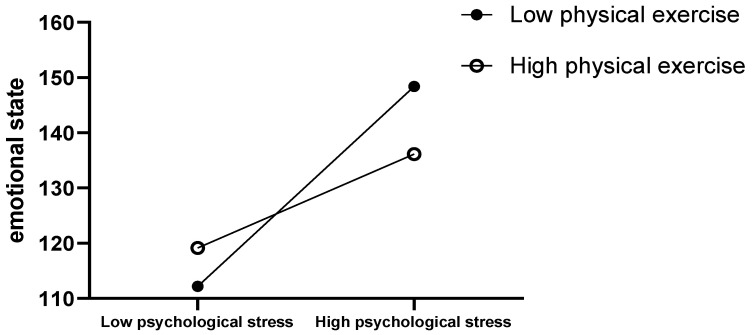
The regulating effect of physical exercise.

**Table 1 healthcare-11-00695-t001:** Descriptive statistics of each variable.

Sex	Age (M ± SD)	Physical Exercise (M ± SD)	Emotional State (M ± SD)	Psychological Stress (M ± SD)
Male (208)	19.371 ± 1.139	27.269 ± 20.255	129.653 ± 24.479	51.170 ± 9.421
Female (286)	19.199 ± 0.997	27.416 ± 22.958	131.653 ± 26.155	49.149 ± 10.335
Total (494)	19.271 ± 1.062	27.354 ± 21.839	130.653 ± 25.263	49.999 ± 10.001

**Table 2 healthcare-11-00695-t002:** Correlation analysis.

	n	Pearson’s r	*p*	Lower 95% CI	Upper 95% CI
PE-PSS	494	−0.637	<0.001	−0.684	−0.585
PE-TMD	494	−0.32	<0.001	−0.387	−0.255
PSS-TMD	494	0.51	<0.001	0.446	0.573

Note: PE = physical exercise, PSS = psychological stress, TMD = emotional state. r is Pearson product difference correlation coefficient.

**Table 3 healthcare-11-00695-t003:** Moderation effect analysis.

Independent Variable	Emotional State			Emotional State		
	*B*	*SE*	*t*	*B*	*SE*	*t*
Intercept	130.654	0.98	133.368 ***	128.979	1.236	104.353 ***
Psychological stress	1.302	0.127	10.236 ***	1.3	0.127	10.263 ***
Physical exercise	0.01	0.058	0.163 **	−0.06	0.066	−0.907 *
Psychological stress × physical exercise				−0.012	0.005	−2.208 **
*R* ^2^	0.26			0.267		
*F*	86.338			59.639		
Δ*R*^2^				0.007		
Δ*F*				4.876		

Note: * *p* < 0.05, ** *p* < 0.01, *** *p* < 0.001.

## Data Availability

Please contact the corresponding authors to obtain the data.

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
