# Peer review of "The Relationship between Psychological Stress and Emotional State in Chinese University Students during COVID-19: The Moderating Role of Physical Exercise"

_healthcare, 2023, doi:10.3390/healthcare11050695_

Round 1

Reviewer 1 Report

Use consistent terminology: Throughout the abstract and the main text, multiple expressions regarding a single concept, such as emotional state, mood state, state of mind, and emotional health, are used.

Abstract

-      On line 1 of the abstract: it says the sample were “randomly selected”; whereas the main text claims that “convenience sampling was adopted.”

-      On line 10 of the abstract: ” ; physical exercise” seems to be a typo?

-      Conclusion: Describe the results reflecting the purpose of study (suggested at the end of the introduction)

2.1. Research object

  - Change the term “object” to “subjects”

- Provide more detailed explanation of the “convenience sampling” method

-      Provide the basis for distributing 715 questionnaires

-      Specify the data collection period

-      What is the meaning of “The selection of all items in the same scale is consistent.”?

-      In “494 questionnaires were finally collected”, wouldn’t “selected” be more adequate than “collected”?

-      Provide the ethical approval code from “Ethics Review Committee of Yangzhou University”

-      Following was not addressed further in the study; consider excluding this part:

“the evaluation standard of exercise amount: ≤19 is classified as small exercise amount; 20 ~ 42 were classified as moderate exercise; ≥43 is classified as a large amount of exercise.”

-      The mood state Scale (POMS): 1) What is the unabbreviated form of POMS, and 2) who developed the scale before it was revised?

2.2 Data Collection and Tools

- The following sentence does not make sense:

“Use 0(none)~4(very much)5 points. total mood disturbance (TMD)= the sum of the scores of each negative emotion subtracting the sum of the scores of each positive emotion disturbance plus 100; the higher the score is, the worse the emotional state is.”

-      Was emotional state subdivided into two variables of negative emotional state and positive emotional state to conduct analysis? If not, these terms seem to be inconsistently used throughout Discussion and Conclusions parts.

-      Also, if two variables were used to conduct analysis, Cronbach α should be provided for each variable.

2.3. statistical approach

- correlation analysis for questionnaire reliability and validity tests: Specify which variables were examined in the correlation analysis

- Elaborate the method for analyzing the moderating effect

- Explicitly propose the research hypotheses

3. Results

- Figure 1. mood state (TMD): State the unabbreviated form for the term

3.2. The regulating effect of physical exercise

- What exactly is the negative emotional state?

- The first sentence is unnecessary

- FIG. 2 The regulating effect of physical exercise: Why was POMS used here?

4. Discussion

- Move the very first paragraph (“This study … among college students.”) into 4.1. Relationship between psychosocial stress and emotional state

- 4.2. The regulating effect of physical exercise: The first paragraph contains the contents irrelevant to the Discussions (such should be dealt in either background or literature parts)

5. Conclusion: Describe the results reflecting the purpose of study (suggested at the end of the introduction)

Reviewer 2 Report

Title
The relationship between psychosocial stress and the emotional state of university students - the moderating role of physical exercise
Comments for the Authors
This study examines the longitudinal relationship between psychosocial stress, emotional status, and physical exercise in university students. Further, if exercise moderates the relationship between stress and emotions. The authors report a negative correlation between exercise and stress, as well as exercise and emotions. The authors also report a positive correlation between stress and emotions. Finally, the authors note that physical exercise moderates the relationship between stress and emotions; this means that exercise helps those with high stress maintain a better or more positive mood state. I commend the authors for this work, but several issues with this version of the manuscript dampen my enthusiasm. Please see below:
Major Issues
1. There are several grammatical, typos, etc., throughout the manuscript; it is beyond my capacity to identify all of them, but I provide some examples below:
a. 2.2 last paragraph: “ina.”
b. 2.3 statistical approach: not capitalized
c. 4.1: “Shouting” is capitalized
2. Unfortunately, I am unsure what is novel about the study. Put simply, a link between stress, mood, and physical exercise appears to be well-established. Are the findings in university students from China the novel element? Or the moderation analysis? Regardless, the reader would benefit significantly from clearly stating the novelty of the study.
3. Extending from comment 2, the title made me believe a moderation analysis was conducted. However, section 3.2 uses the term “adjustment.” Perhaps, this is a type of moderation analysis and reflects my lack of statistical expertise, but this should be clarified.
4. The Discussion highlights the influence of the COVID-19 pandemic. If the goal of this manuscript was to assess during the pandemic, then this should be clarified. Regardless, the level of lockdown or restrictions during data collection should be clearly stated, given the variables of interest.
Minor Issues
Title & Abstract
1. Should clarify that the results are from a single university in China.
Introduction
1. Several sentences require references.
2. Overall, it is disjointed and is missing critical information while also providing unnecessary information. For example, given the journal's broad readership, it would be worth explaining physical exercise and how it differs from physical activity, two overlapping but distinct concepts.
Methods
1. Use the term gender but only report males and females. According to CIHR guidelines, males and females would be considered sex and not gender.
2. Exercise was tested using a physical activity scale. Exercise and physical activity are two overlapping but distinct concepts.
3. Scoring for exercise and mood state needs to be clarified. Would benefit from a clear formula and an example.
Results
1. No table of participant characteristics.
2. Most of section 3.2 belongs in Methods.
Discussion
1. Most of what I said about the Introduction also applies here.
2. Defines psychosocial stress at the start of the first paragraph in sections 4.1 and 4.2.
3. Missing a section dedicated to Limitations & Future Directions.
4. I did like the Cross-stress hypothesis.
Tables & Figure
1. Figures are too small and, therefore, unclear.
2. Figures need to stand on their own, with a more detailed explanation.
3. Abbreviations change. For example, mood goes from “POMS” to “TMD.”

Reviewer 3 Report

Congratulations to the authors for the article. The approach of the article is interesting, but some improvements are needed to make it publishable.

It is striking that appropriate statistical tests are not used for the medication model, e.g. according to Hayes. The description of the statistical section is too short.

I do not think that figures for the correlations are necessary, a table is sufficient.

Using for example a picture to show the mediation model stated in the objective would help a lot to understand it better.

It would be convenient to always use the same terms and acronyms for the study variables, as this leads to confusion.

In the discussion, many quotations are used that have not been used in the introduction. The introduction is somewhat sparse in terms of the bibliography used.

In the conclusion there is no reference to mediation. If it appears in the title of the article, it is supposed to be relevant enough to be referred to in the conclusion.

Round 2

Reviewer 3 Report

Congratulations to the authors on the improvement of the article.

Author Response

Thank you very much for your valuable suggestions. Your suggestions are a significant improvement to our article.